# Super-resolution imaging uncovers the nanoscopic segregation of polarity proteins in epithelia

**Pierre Mangeol[1]\*, Dominique Massey-Harroche[1], Fabrice Richard[1], Jean-Paul Concordet[2], Pierre-François Lenne[1]\*†, André Le Bivic[1]\*†**

[1]Aix-Marseille University, CNRS, UMR7288, Developmental Biology Institute of Marseille (IBDM), Marseille, France; [2]Laboratoire Structure et Instabilité des Génomes, Muséum National d'Histoire Naturelle (MNHN), Institut National de la Santé et de la Recherche Médicale (INSERM), U1154, Centre National de la Recherche Scientifique (CNRS), Paris, France

**Abstract** Epithelial tissues acquire their integrity and function through the apico-basal polarization of their constituent cells. Proteins of the PAR and Crumbs complexes are pivotal to epithelial polarization, but the mechanistic understanding of polarization is challenging to reach, largely because numerous potential interactions between these proteins and others have been found, without a clear hierarchy in importance. We identify the regionalized and segregated organization of members of the PAR and Crumbs complexes at epithelial apical junctions by imaging endogenous proteins using stimulated-emission-depletion microscopy on Caco-2 cells, and human and murine intestinal samples. Proteins organize in submicrometric clusters, with PAR3 overlapping with the tight junction (TJ) while PALS1-PATJ and aPKC-PAR6β form segregated clusters that are apical of the TJ and present in an alternated pattern related to actin organization. CRB3A is also apical of the TJ and partially overlaps with other polarity proteins. Of the numerous potential interactions identified between polarity proteins, only PALS1-PATJ and aPKC-PAR6β are spatially relevant in the junctional area of mature epithelial cells, simplifying our view of how polarity proteins could cooperate to drive and maintain cell polarity.

**\*For correspondence:**
pierre.mangeol@univ-amu.fr (PM);
pierre-francois.lenne@univ-amu.fr (P-FL);
andre.le-bivic@univ-amu.fr (ALB)

†These authors contributed equally to this work

**Competing interest:** The authors declare that no competing interests exist.

## Editor's evaluation

This important work advances our understanding of how apical polarity proteins are organized in epithelia using advanced, super-resolution imaging. The microscopy performed in this study is skillful and beautifully presented and clarifies the view of how polarity proteins may interact in mature epithelia. This article will be of interest to cell biologists, especially those interested in cell polarity and tissue architecture.

## Introduction

In epithelial tissues, cells coordinate their organization into a polarized sheet of cells. Each cell acquires an apico-basal organization and specialized lateral junctions, namely, tight junctions (TJs, also known as zonula occludens), adherens junctions, and desmosomes (*Farquhar and Palade, 1963*). This organization is key to the development, maintenance, and function of epithelial tissues.

Over the past two decades, several proteins have been discovered to be pivotal to epithelial polarization, such as PAR3, PAR6, aPKC (PAR complex), Crumbs, PATJ, PALS1 (Crumbs complex), Scribble, LGL, and DLG (Scribble complex) in mammals (for review, see *Assémat et al., 2008*; *Pickett et al.,*

**eLife digest** Many of our organs, including the lungs and the intestine, are lined with a single layer of cells that separate the inside of the organ from the surrounding environment inside the body. These so-called epithelial cells form a tightly packed barrier and have a very characteristic organization.

The apical surface faces the outside world, while the basal surface faces the inner tissues. These different interfaces are reflected in the organization of the cells themselves. The shape, composition, and role of the apical cell surface are distinct from those of the basal surface, and they also contain different proteins. In some epithelial cells, the apical surface specializes and forms protruding structures called microvilli. Thus, epithelial cells are said to be polarized along this apical–basal axis. Over the last 30 years, many labs have identified and studied which proteins help epithelial cells become and stay polarized.

Previous biochemical experiments showed that these so-called polarity proteins interact with each other in many different ways. But it remains unclear whether some of these interactions are more important than others, and where exactly in the apical or basal membranes these interactions take place.

Mangeol et al. used super-resolution microscopy to observe the polarity of proteins at the apical membranes of both human and mouse cells from the small intestine to answer these questions. They focused on areas called tight junctions, where the intestinal cells connect with each other to form the barrier between the outside and the inside. First, all the polarity proteins clustered together in various formations, they were not distributed uniformly. For example, one protein called PAR3 was at the level of the tight junctions, whereas other proteins were closer to the apical surface and the outside world. Only two pairs of proteins – PAR6 and aPKC, and PALS1 and PATJ – formed stable clusters with each other. This finding was unexpected because previous biochemical experiments had predicted multiple interactions. Third, the PALS1/PATJ complexes stayed at the bottom of the microvilli protrusions, whereas PAR6/aPKC were inside the protrusions.

Taken together, these experiments reveal a detailed snapshot of how the polarity proteins themselves are organized at the apical surface of epithelial cells. Future work will be able to address how these protein complexes behave over time.

*2019*; *Rodriguez-Boulan and Macara, 2014*). These proteins are remarkably well conserved over the animal kingdom (*Belahbib et al., 2018*; *Le Bivic, 2013*). Deletion or depletion of one of these proteins usually results in dramatic developmental defects (*Alarcon, 2010*; *Charrier et al., 2015*; *Hakanen et al., 2019*; *Lalli, 2012*; *Park et al., 2011*; *Sabherwal and Papalopulu, 2012*; *Tait et al., 2020*; *Whiteman et al., 2014*).

In the quest to understand the role of polarity proteins, numerous genetic and biochemical studies have been carried out. We and others have found that these proteins interact to form multiprotein complexes. Pioneering studies defined three core complexes based on the discovery of protein interactions or localization: the PAR complex consisting of PAR3, PAR6, and aPKC proteins (*Joberty et al., 2000*; *Lin et al., 2000*), the Crumbs complex consisting of CRUMBS, PALS1, and PATJ (*Bhat et al., 1999*; *Makarova et al., 2003*; *Roh et al., 2002b*), and the Scribble complex consisting of Scribble, LGL, and DLG (*Bilder et al., 2000*). However, this view became more complex over the years as many interactions between proteins of different complexes can occur (*Assémat et al., 2008*; *Hurd et al., 2003*; *Lemmers et al., 2004*), and interactions of polarity proteins with cytoskeleton regulators and lateral junction proteins are common (*Assémat et al., 2008*; *Chen and Macara, 2005*; *Itoh et al., 2001*; *Médina et al., 2002*; *Michel et al., 2005*; *Roh et al., 2002a*; *Takekuni et al., 2003*; *Tan et al., 2020*). A current limitation in the understanding of polarization is that there is no clear hierarchy of the importance of these numerous interactions. Potential interactions revealed through biochemical assays do not necessarily reflect relevant interactions in cells and do not specify when nor where in the cell these interactions could be relevant.

Polarity proteins have been localized with classical light microscopy, and remarkably, they are often found concentrated at the apical junction, a key organizational landmark of epithelial cells. To understand how polarity proteins cooperate to orchestrate cell polarization, one needs to understand how precisely polarity proteins organize with respect to apical junctions or the cytoskeleton. However,

except for a few limited cases (*Hirose et al., 2002*; *Izumi et al., 1998*; *Tan et al., 2020*), the precise localization of polarity proteins at these organizational landmarks is missing. Moreover, knowing how polarity proteins organize in relation to each other in the cell should enable us to decipher from their plentiful known potential interactions, which ones are more relevant in specific subregions of the cell.

To tackle these challenges, we decided to systematically localize with stimulated-emission-depletion (STED) microscopy the polarity proteins that are key to the establishment of the apical pole of epithelia: PAR3, aPKC, PAR6β, PATJ, PALS1, and CRB3A. These proteins localize at the apical junction region of epithelial cells. Because how proteins interact and localize is likely to depend on cell differentiation, we decided to focus here on mature epithelia, a state where we hypothesize that protein interactions and localization are stationary. Using human and murine intestine and Caco-2 cells, we first imaged endogenous polarity proteins with respect to the TJ to appreciate their overall organization in the region. Second, we localized these proteins two-by-two to uncover relevant apical polarity protein subcellular associations. Finally, we focused on the organization of polarity proteins with respect to the actin cytoskeleton. We find that polarity proteins localize in distinct subregions that do not reflect the canonical definition of polarity protein complexes. In addition, their localization with respect to the cytoskeleton emphasizes some emerging roles of polarity proteins as regulators of actin organization.

## Results

### Polarity proteins are localized in separate subdomains in the apical junction region

To obtain a first estimate of polarity protein localization in the TJ region, we systematically imaged each polarity protein with respect to a marker of the TJ. To this end, each apical polarity protein and a TJ marker (ZO-1 or Occludin) were immunostained and imaged together using STED microscopy (*Hell and Wichmann, 1994*; *Figures 1 and 2*, *Figure 1—figure supplement 1*, *Figure 2—figure supplement 1*). STED images were acquired in the TJ region both in the apico-basal and the planar orientations of cells in human and mouse intestinal biopsies (*Figure 1*) and Caco-2 cells (*Figure 2*). To optimize the sample orientation, samples were cryo-sectioned when needed, in particular to obtain apico-basal orientation. Since we focused on mature epithelia, intestinal cells were observed exclusively in villi and Caco-2 cells were seeded on filters and grown over 14 days to allow sufficient differentiation (*Pinto et al., 1983*). Because the resolution of STED microscopy followed by deconvolution was, in our hands, about 80 nm in each color channel, the gain of resolution compared to classical confocal microscopy approaches was threefold in the planar orientation and sevenfold along the apico-basal axis.

We found that the localization of each polarity protein was conserved across all samples and species (*Figures 1 and 2*). All proteins were concentrated in the TJ region as clusters of typically 80–200 nm in size (the smallest cluster size found is likely due to the imaging resolution limit), but their precise localization was protein-dependent. We could group proteins into three main localization types. While we mostly found PAR3 at the TJs (*Figures 1A, C, D, and F and 2A and C*), PAR6β and aPKC were at the TJ level and apical of the TJ (*Figures 1A and C and 2A and C*). We found CRB3A, PALS1, and PATJ almost exclusively apical of the TJ (*Figures 1A, C, D, and F and 2A and C*). Interestingly, we often found PAR6β, aPKC, CRB3A, PALS1, and PATJ separated laterally from the TJ since we frequently detected clusters of these proteins 100–200 nm away from the TJ (*Figures 1A, B, D, and E and 2A and E*). There were some slight differences between intestinal samples and Caco-2 cells; in particular, CRB3A was more spread from the apical to the TJ domain in Caco-2 cells (*Figure 2C*). These differences may originate from sample preparation or differences in cell organization due to tissue maturation. These first results show that polarity proteins organize in separate subdomains in the TJ region, namely, PAR3 at the TJ, and the other polarity proteins studied mostly apical of the TJ.

### Confirmation of the cluster organization by alternative methods

The observation that polarity proteins can organize as clusters has been reported in *Caenorhabditis elegans* (*Dickinson et al., 2017*; *Wang et al., 2017*), but it is still possible that in our case clustering is an artifact of antibody staining. Antibody staining is known to generate artifacts because of permeabilization, fixation, and antibody specificity (*Schnell et al., 2012*), even though a comparison between

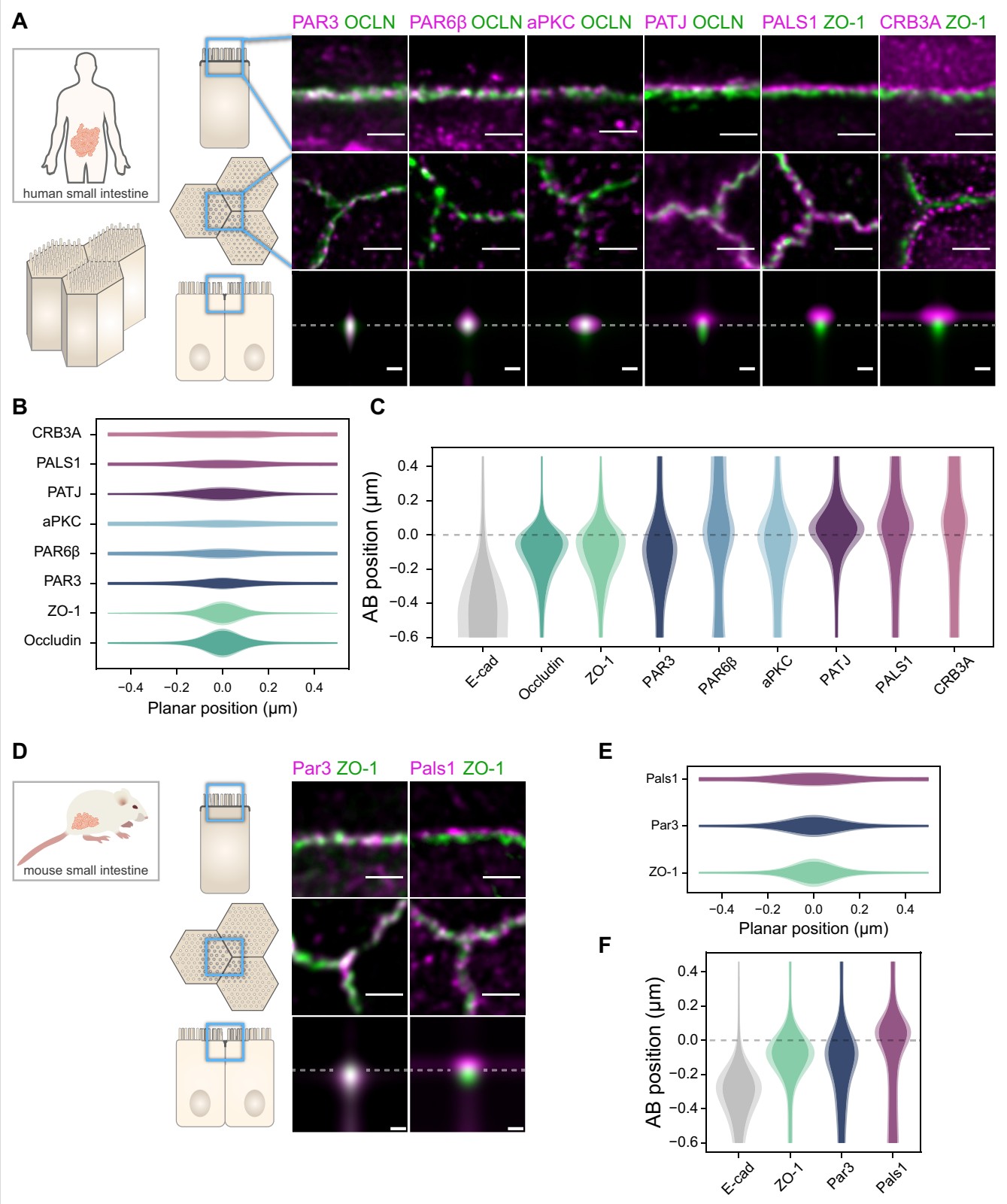

**Figure 1.** Polarity proteins localize in separate subdomains in the tight junction (TJ) region in human (**A–C**) and murine (**D–F**) small intestine biopsies. (**A, D**) STED images of protein localization in the TJ area. TJ proteins in green, polarity proteins in magenta. Top row: apico-basal orientation; middle row: planar orientation; bottom row: estimates of average protein localization in the apico-basal orientation perpendicular to the junction, obtained by multiplying average localizations estimated in (**B**) and (**C**) for human biopsies and (**E**) and (**F**) for murine biopsies. Top row and middle row, scale bar

*Figure 1 continued on next page*

*Figure 1 continued*

1 µm; bottom row, scale bar 200 nm. (**B, E**) Average localization of polarity proteins in the planar orientation, obtained by measuring the intensity profile of proteins perpendicular to the junction, using the TJ protein position as a reference. (**C, F**) Average localization of polarity proteins in the apico-basal orientation, obtained by measuring the intensity profile of proteins along the apico-basal orientation, using the TJ protein position as a reference. In (**B, C, E, F**), on a given position dark colors represent average intensity values, and lighter colors the average added with the standard deviation. We used three biological replicates for each human and mouse experiment (details in *Figure 1—source data 1*). Details of the analysis are specified in the 'Materials and methods' section.

The online version of this article includes the following source data and figure supplement(s) for figure 1:

**Source data 1.** Details of the number of junctions used in each replicate.

**Source data 2.** Estimate of the average density versus localization for each protein.

**Figure supplement 1.** Localization of ZO-1 vs. Occludin in the human small intestine and E-cadherin vs. ZO-1 in human and mouse small intestine.

**Figure supplement 1—source data 1.** Details of the number of junctions used in each replicate.

**Figure supplement 1—source data 2.** Estimate of the average density versus localization for each protein.

---

antibody staining and fluorescence tagging in mammalian cells has found very good agreement when using confocal microscopy (*Stadler et al., 2013*). We used two different approaches to assess whether protein clusters were due to our labeling protocol.

First, we prepared a CRISPR-Cas9 knock-in Caco-2 cell line expressing PAR6β tagged with the fluorescent protein Citrine (see 'Materials and methods' for preparation details). To compare how proteins localized when imaged with their fluorescent tag or with antibody staining, we achieved the following: we first imaged live PAR6β tagged with Citrine with STED, then fixed, permeabilized, and labeled the cells with antibodies against PAR6β. We finally came back to the same cells observed live and imaged them with STED, this time using Alexa Fluor-568 conjugated antibody. We find a very good agreement between images observed live with Citrine or fixed with Alexa Fluor-568 (*Figure 3A*). In particular, we find that PAR6β organizes in clusters in live images, and most clusters found with live imaging can be found again with immunolabeling.

Second, to further examine whether permeabilization with detergent can generate clusters, we chose to permeabilize fixed cells by freezing them in liquid nitrogen and thawing them at room temperature. Such treatment is known to destabilize cell membranes (*Steponkus and Lynch, 1989*). After this treatment, we labeled cells with antibodies against PATJ and observed the preparation with STED (*Figure 3B*). Images obtained with this protocol or the one using detergent gave comparable results. Clusters of proteins can also be observed without the use of detergent.

We conclude that the clustered organization of proteins we observe is not due to methodological artifacts as we can find the same clusters of PAR6β proteins in live and fixed CRISPR-Cas9 knock-in Caco-2 cells, and the absence of detergent when staining cells with antibodies against PATJ does lead to the same organization as the one observed with detergent.

## Redefining relevant interactions between polarity proteins from colocalization analysis

The organization of proteins in separate subdomains led us to investigate how polarity proteins were organized within these subdomains, and more specifically how clusters of polarity proteins were localized with respect to each other. To tackle this question, we imaged polarity proteins two-by-two in Caco-2 cells and quantified the extent of their colocalization using the protein–protein proximity index developed in *Wu et al., 2010*, providing a quantitative estimate of protein proximity (*Figure 4*). Because of the organization of protein clusters, different proteins that localize at the same level on the apico-basal axis may appear as overlapping 'more' when observed in the apico-basal orientation rather than when they are observed in the planar orientation (*Figure 4—figure supplement 1*); this is because the axial resolution (about 550 nm) is sevenfold lower than the planar resolution (about 80 nm). To circumvent this limitation, we minimized the apparent colocalization for each protein pair by orienting our sample either in the planar or apico-basal orientation, wherever apparent colocalization was lowest.

First, we found that some of the proteins colocalize strongly: PALS1 with PATJ and aPKC with PAR6β, presumably in both cases forming a complex, as the literature suggests (*Joberty et al., 2000*;

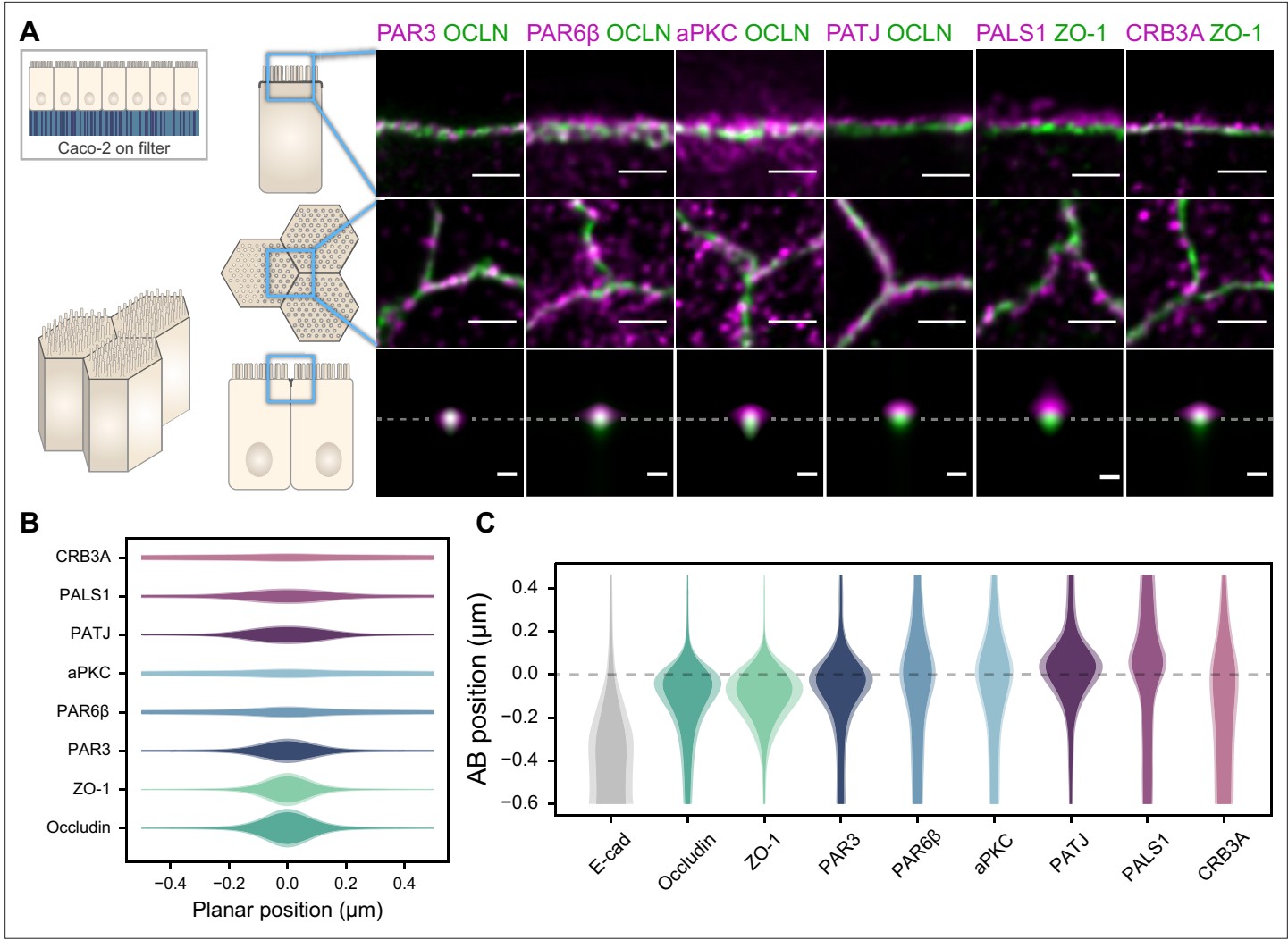

**Figure 2.** Polarity proteins localize in separate subdomains in the tight junction (TJ) region in Caco-2 cells. (**A**) Stimulated-emission-depletion (STED) images of protein localization in the TJ area. TJ proteins in green, polarity proteins in magenta. Top row: apico-basal orientation (obtained from cryo-sectioning cells grown on filter); middle row: planar orientation; bottom row: estimates of average protein localization in the apico-basal orientation perpendicular to the junction, obtained by multiplying average localizations estimated in (**B**) and (**C**). Top row and middle row, scale bar 1 μm; bottom row, scale bar 200 nm. (**B**) Average localization of polarity proteins in the planar orientation obtained by measuring the intensity profile of proteins perpendicular to the junction, using the TJ protein position as a reference. (**C**) Average localization of polarity proteins in the apico-basal orientation obtained by measuring the intensity profile of proteins along the apico-basal orientation, using the TJ protein position as a reference. In (**B, C**), on a given position dark colors represent average intensity values, and lighter colors the average added with the standard deviation. We used three cell culture replicates (details in *Figure 1—source data 1*). Details of the analysis are specified in the 'Materials and methods' section.

The online version of this article includes the following source data and figure supplement(s) for figure 2:

**Source data 1.** Details of the number of junction used in each replicate.

**Source data 2.** Estimate of the average density versus localization for each protein.

**Figure supplement 1.** Localization of ZO-1, Occludin, and E-cadherin ZO-1 Caco-2 cells.

**Figure supplement 1—source data 1.** Details of the number of junction for each replicate.

**Figure supplement 1—source data 2.** Estimate of the average density versus localization for each protein.

*Lin et al., 2000*; *Roh et al., 2002b*; *Figure 4A*). Surprisingly, we found PALS1-PATJ and aPKC-PAR6β well segregated from each other when we observed them in the planar orientation. They sometimes appeared as alternating bands along the junction with a spatial repeat in the range of 200–300 nm (zooms in *Figure 4A*). In some cases, these bands seemed formed by clusters facing each other in neighboring cells, indicating potential coordination of polarity protein organization between adjacent

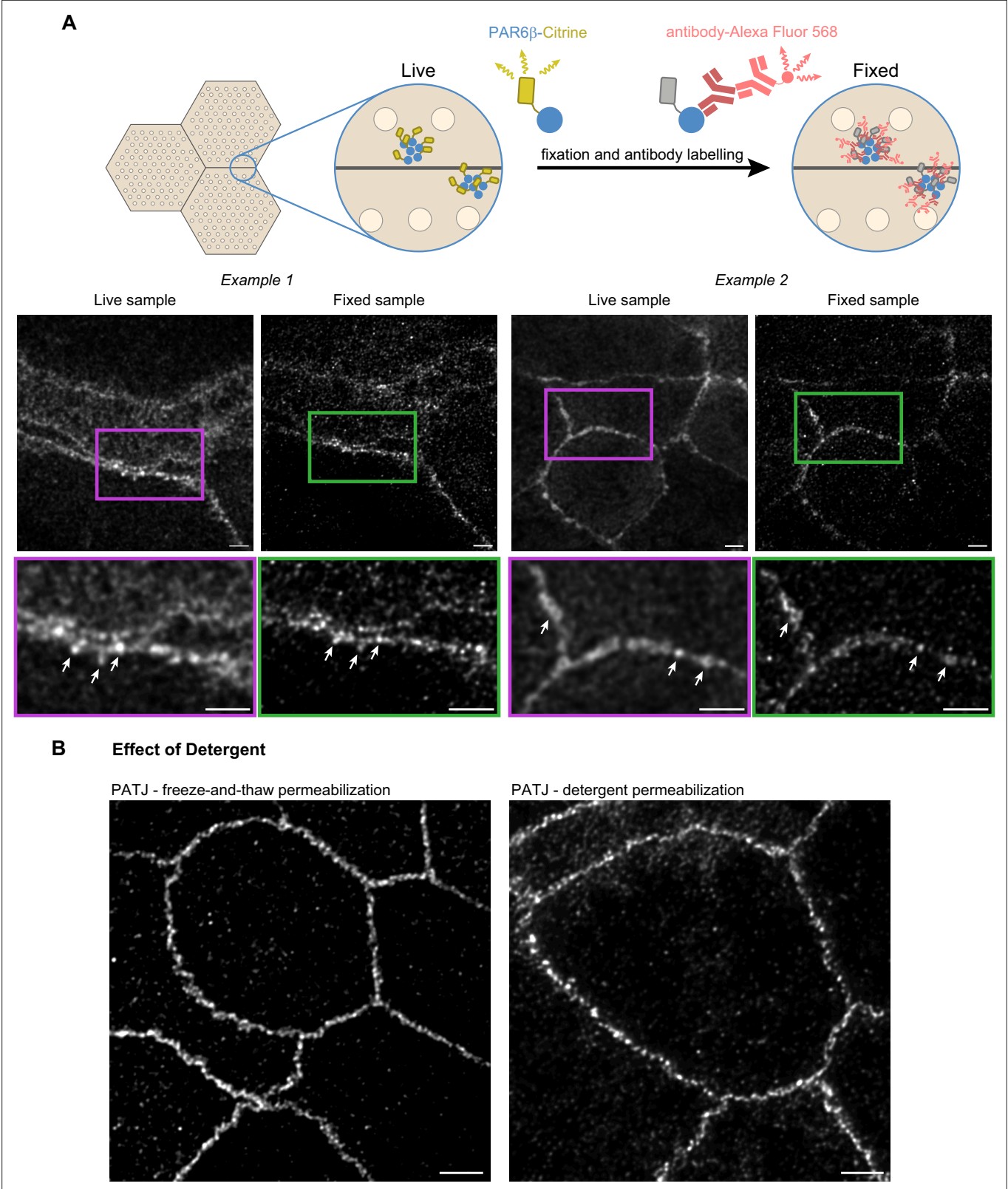

**Figure 3.** Confirmation of the cluster organization by alternative methods. (**A**) Two examples of stimulated-emission-depletion (STED) images obtained on living Caco-2 cells expressing PAR6β-Citrine that were then fixed and immunolabeled and zoom on junctions (insets). Imaging of the same cells shows that clusters are observed in live and fixed conditions (arrows pointing at the same clusters in both conditions). (**B**) Images showing that permeabilization using freeze-and-thaw or detergent lead to very similar results, showing that detergents are not the cause of protein clustering. Scale bars: 2 µm. We obtained the same conclusions on three independent cell culture replicates.

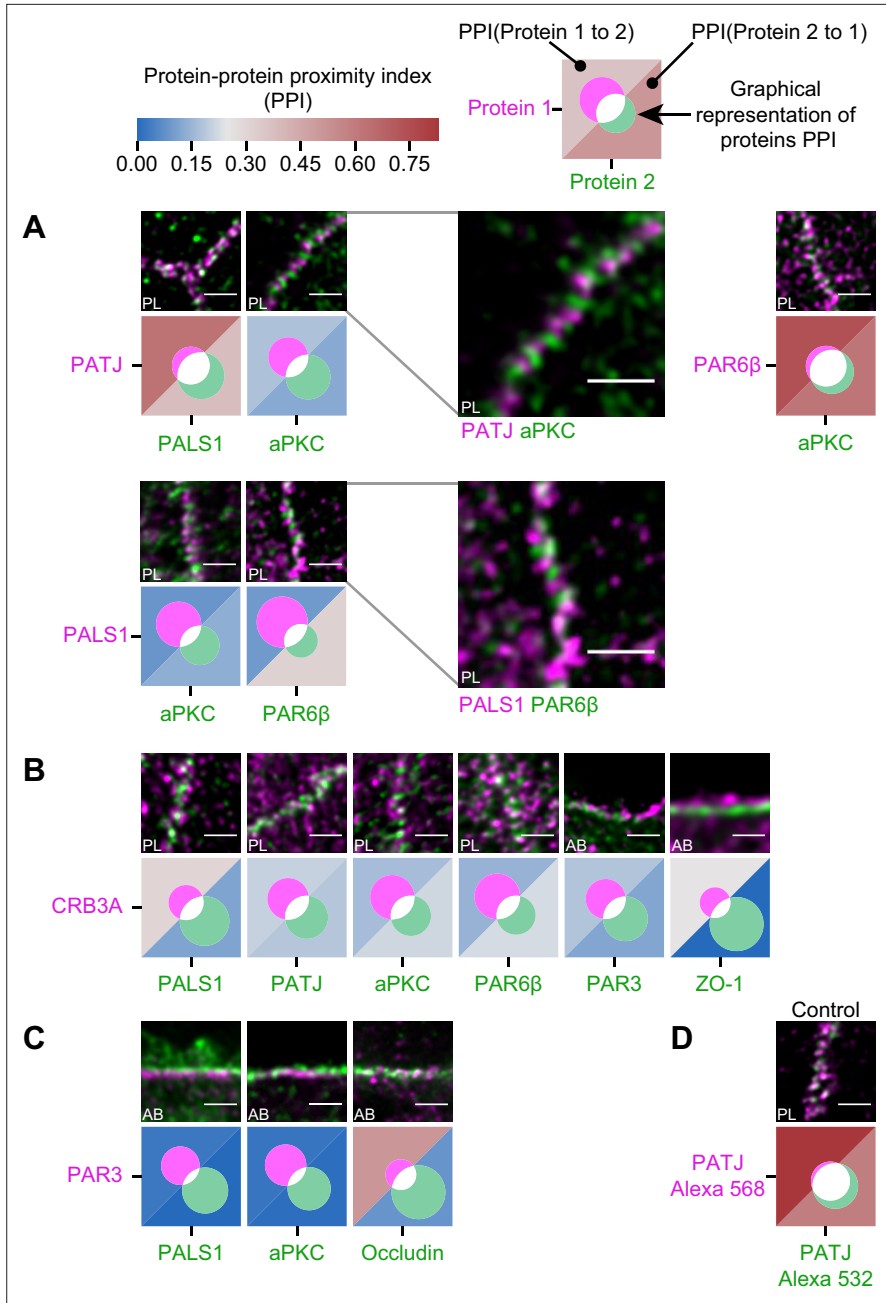

**Figure 4.** Proximity analysis of polarity proteins redefines protein complexes. The analysis is carried out in Caco-2 cells, where we used the concept of protein–protein proximity index (PPI) introduced by *Wu et al., 2010*, indicating the proximity of two different protein populations. PPI of 0 indicates no proximity (or no colocalization), and PPI of 1 indicates perfect proximity (or perfect colocalization); intermediate values give an estimate of the fraction of a given protein being in close proximity (or colocalize) with another one. Here, the result of the proximity analysis is represented graphically with color-coded values and Venn diagrams as depicted on the top of the figure (details in 'Materials and methods'). The analysis has been carried out on apico-basal (AB) or planar (PL) orientation images to minimize apparent colocalization due to overlapping in different planes; this is reported in the representative image of each experiment. (**A**) Proximity analysis for PATJ, PALS1, aPKC, and PAR6β and corresponding representative images. Zoomed images (PATJ/aPKC and PALS1/PAR6β) illustrate the segregation of these proteins. (**B**) Proximity analysis for CRB3A and the other polarity proteins. (**C**) Proximity analysis for PAR3 with PALS1, aPKC, and Occludin. (**D**) Control experiment with PATJ labeled with an Alexa 532 secondary antibody and an Alexa 568 tertiary antibody. We used three cell culture replicates for each protein pair (details in *Figure 4— source data 1*). The details of the analysis are specified in 'Materials and methods.' Scale bars: 1 μm.

*Figure 4 continued on next page*

*Figure 4 continued*

The online version of this article includes the following source data and figure supplement(s) for figure 4:

**Source data 1.** Details of the number of junctions for each repicate.

**Source data 2.** PPI index for each protein couple.

**Figure supplement 1.** Protein–protein proximity index (PPI) analysis on the orientations can lead to artificially higher PPI because of higher protein overlap.

**Figure supplement 1—source data 1.** Details of the number of junctions for each replicate.

cells. Second, we found that only a minority of CRB3A colocalized with any of the other polarity proteins (*Figure 4B*). These observations are also surprising because CRB3A has been reported to strongly interact both with PALS1 and PAR6 (*Hayase et al., 2013*; *Lemmers et al., 2004*; *Li et al., 2014*; *Makarova et al., 2003*). This could mean that these interactions are mostly transient or that they are not prominent in the TJ area. This result questions the stability and functional cellular meaning of the canonical CRB3-PALS1-PATJ complex and the CRB3-PAR6 interaction. Finally, when localizing PAR3 along with PALS1 or aPKC, we found that PAR3 is hardly found with either of these proteins (*Figure 4C*). These data show that PAR3, aPKC, and PAR6β do not associate in a static complex as has been suggested in several non-mammalian models (*Afonso and Henrique, 2006*; *Harris and Peifer, 2005*; *Morais-de-Sá et al., 2010*; *Rodriguez et al., 2017*). It appears, in our conditions, that aPKC and PAR6β are likely linked in the apical TJ region, whereas PAR3 is poorly associated with them. Again, it is possible that the interaction between PAR3 and PAR6β-aPKC is mostly transient or that it is not relevant in the TJ area. We conclude that PAR3 is largely excluded from other polarity proteins at the TJ and that PALS1-PATJ, PAR6β-aPKC, and CRB3 form three spatially separated entities in the apical region of the TJ.

## PATJ localization in the TJ region with electron tomography

In the generally accepted description of the canonical Crumbs complex, PALS1 binds to the transmembrane protein CRB3A and PATJ binds to PALS1 (*Roh et al., 2002b*). Therefore, PALS1 and PATJ are thought to be in close vicinity of the membrane since CRB3A is a short transmembrane protein. Moreover, it was proposed that PATJ links CRB3A-PALS1 to the TJ area (*Michel et al., 2005*) because of the direct interaction of PATJ with the TJ protein ZO-3 and Claudin1 (*Roh et al., 2002a*). Our protein-proximity analyses, however, raise the question of whether PALS1/PATJ interact with CRB3A in the TJ region (*Figure 4*), and our localization of PATJ with STED suggests that most PATJ proteins are often too far from the TJ to interact with this structure (*Figures 1 and 2*). Therefore, to obtain a more complete understanding of PATJ localization in the TJ region, we observed PATJ with electron tomography using immunogold labeling in Caco-2 cells (*Figure 5*).

Consistent with what we observed with STED, we often found PATJ organized in clusters apical of the TJ (*Figure 5A*). We started by quantifying PATJ position with respect to the TJ, using as a reference the most apical part of the TJ (defined morphologically as the most apical position of contact between neighboring cells' plasma membranes) (*Figure 5B*). We found that most PATJ proteins were about 80 nm away from the TJ (*Figure 5C*). Although PATJ molecular structure is not known, given its sequence including multiple potent unstructured domains between PDZ domains, it is likely that as a folded protein its size cannot fill the 80 nm gap we find. Therefore, our data suggest that most PATJ molecules do not interact directly with TJ proteins. We found instead most PATJ proteins were close to the apical membrane and that only a small fraction was present in microvilli or the cytoplasm (*Figure 5D*). Previous observations that PATJ associates with ZO-3 or Claudin1 might depend on the cellular state, or these interactions could be transient.

CRB3A is thought to anchor PALS1 and PATJ to the plasma membrane. However, given our results showing a minor colocalization of PATJ and PALS1 with CRB3A, it is unlikely to be the case for most PALS1 and PATJ molecules. Therefore, the localization of PATJ close to the apical membrane led us to wonder whether PATJ together with PALS1 could be associated with the apical plasma membrane via interactors that remain to be discovered. Thus, we measured the distance of the immunogold label of PATJ to the plasma membrane (*Figure 5E*) and found that the distance of the gold label is in most cases compatible with the association of PATJ and PALS1 with the apical plasma membrane (123/169 ≈ 73% of gold particles were less than 38 nm away from the plasma membrane, corresponding to the

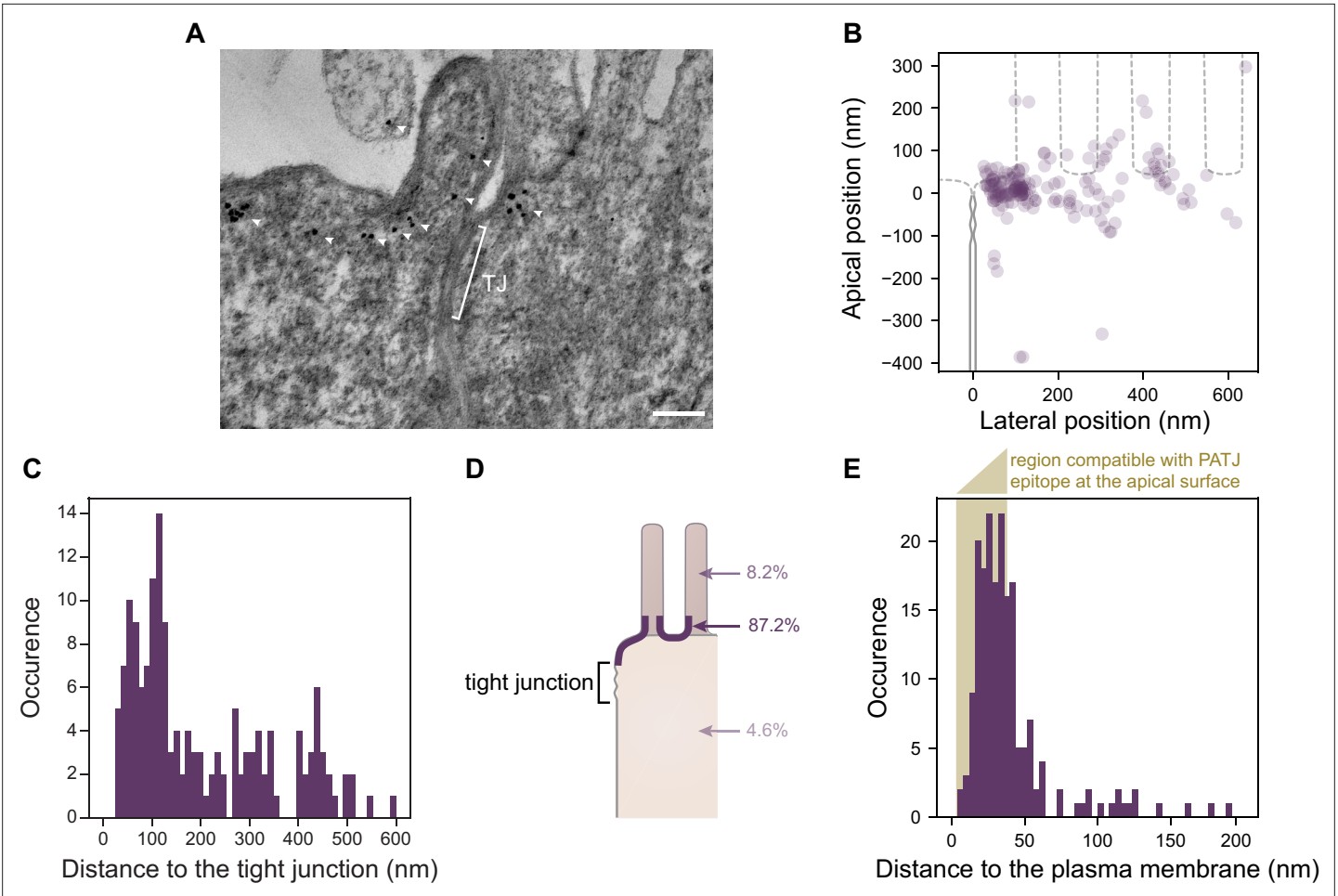

**Figure 5.** Electron tomography shows that PATJ localizes as clusters at the plasma membrane apically of the tight junction (TJ) in Caco-2 cells. (**A**) Representative image of PATJ labeled with gold particles (arrowheads pointing at single particles or clusters of particles). The bracket with TJ indicates the TJ. Minimum intensity projection of a 150-nm-thick tomogram, scale bar: 100 nm. (**B**) Localization of gold particles labeling PATJ with respect to the TJ both in the apico-basal and lateral directions. (**C**) Distance between the center of gold particle labels and the TJ. (**D**) Summary of gold particles localization in the microvilli, in the vicinity of the plasma membrane and the cytoplasm. (**E**) Distance between gold particles and the apical surface. In amber, the region of distances compatible with PATJ epitope being at the apical surface, between 3 nm (radius of gold particles) and 37 nm (size of the primary and gold-labeled secondary antibody combination added with the presumed size of PALS1; *Li et al., 2014*). Tomograms of 300 nm in thickness of 12 junctions were used to extract the position of 169 gold particles labeling PATJ proteins. These junctions were obtained from one cell culture.

The online version of this article includes the following source data for figure 5:

**Source data 1.** Individual distance of gold particles versus the tight junction and the apical membrane.

size of the primary and gold-labeled secondary antibody combination added with the size of PALS1). We conclude that PATJ and PALS1 are likely to be anchored to the apical membrane not by CRB3A but by yet unknown apical membrane proteins.

## Organization of PATJ-PALS1, PAR6β-aPKC, and the actin cytoskeleton

Because polarity proteins play a key role in the epithelial organization, we wondered how these proteins were organized with respect to the actin cytoskeleton. Actin is very densely packed in the TJ region, which makes challenging the identification of where polarity proteins are localized with respect to the actin structure, if not impossible, with 2D STED. Indeed, the method leads to overlaps between actin structures that are then very difficult to extricate. Therefore, we chose to use 3D STED, which has a planar resolution lower than 2D STED (about 120 nm) but a much higher axial one (about

**A**

F-actin PAR6β  F-actin aPKC  F-actin PALS1  F-actin PATJ

PL  PL  PL  PL

AB  AB  AB  AB

**B**

microvilli
microvilli base
microvilli vicinity

**Figure 6.** Organization of PAR6β, aPKC, PATJ, and PALS1, with respect to the actin cytoskeleton. (**A**) 3D stimulated-emission-depletion (STED) imaging of cells labeled with Phalloidin and antibodies against polarity proteins with (top) top view and (bottom) side view on cell–cell junctions. Scale bars: top 2 µm, bottom 1 µm. (**B**) Localization analysis of PAR6β, aPKC, PATJ, and PALS1 vs. microvilli organization. We used three independent cell cultures. Detailed counts of clusters are given in *Figure 6—source data 1*.

The online version of this article includes the following source data for figure 6:

**Source data 1.** Contingency table of the number of counts of polarity proteins in the different microvilli regions.

140 nm), making it a tool of choice to decipher how polarity proteins organize together with actin in three dimensions.

We find that in the region where PATJ, PALS1, PAR6β, and aPKC are located, the actin cytoskeleton identified with 3D STED consists mostly of microvilli structures (*Figure 6A*). PATJ and PALS1 are found mostly at the base of microvilli or in between microvilli (*Figure 6*), in agreement with what we found on PATJ localization with electron microscopy (*Figure 5*). PAR6β and aPKC are mostly found within microvilli or close to the base of microvilli, but hardly between microvilli (*Figure 6*). This result reinforces our previous

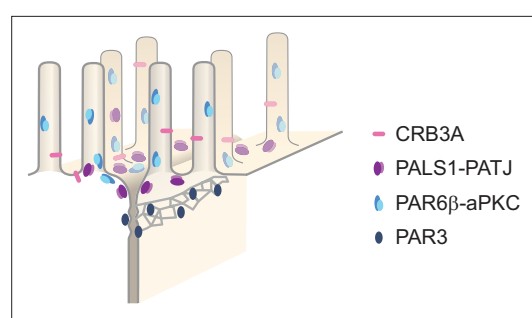

CRB3A
PALS1-PATJ
PAR6β-aPKC
PAR3

**Figure 7.** Organizational model of polarity proteins in the tight junction (TJ) region.

observations that PATJ-PALS1 and PAR6β-aPKC form separate clusters as we find them at different locations within the apical region of the junction.

## Discussion

In this study, we have systematically localized polarity proteins with super-resolution microscopy in epithelial cells. We observed endogenous PAR3, aPKC, PAR6β, PATJ, PALS1, and CRB3A in the human intestine and Caco-2 cells, and PAR3 and PALS1 in the mouse intestine. We found the following (*Figure 7*): (1) all these polarity proteins organize as submicrometric clusters concentrated in the TJ region. PAR3 localizes at the TJ, aPKC, and PAR6β localize at the TJ level, but mostly apically of the TJ, while PATJ, PALS1, and CRB3A are apical of the TJ (*Figures 1 and 2*). (2) PAR6β-aPKC and PATJ-PALS1 form two pairs that are often respectively found in the same clusters (*Figure 4A*), strongly indicating that these respective proteins form a stable and major complex in this region of the cells (i.e., the PAR6-aPKC complex and the PALS1-PATJ complex). (3) Unexpectedly, PALS1-PATJ and PAR6β-aPKC clusters are segregated from each other (*Figure 4A*). Our data show that these clusters are concentrated in the first one or two rows of microvilli, with the PAR6β-aPKC complex localized at the base and within microvilli, whereas the PALS1-PATJ complex is localized at the base and in between these microvilli (*Figures 5 and 6*), further showing that the PAR6β-aPKC complex and the PALS1-PATJ complex are spatially separated in the cell. (4) CRB3 shows little association with any of the other polarity proteins (*Figure 4B*), questioning how PALS1-PATJ and PAR6β-aPKC are mechanistically recruited to the plasma membrane and localized to the apical surface.

Previous studies were largely based on biochemical approaches. The first interactions found defined canonical polarity protein complexes, while subsequent studies highlighted the numerous potential interactions that can be found with such approaches, between polarity proteins of different complexes (*Assémat et al., 2008*; *Bhat et al., 1999*; *Hurd et al., 2003*; *Joberty et al., 2000*; *Lemmers et al., 2004*; *Lin et al., 2000*; *Makarova et al., 2003*; *Roh et al., 2002b*), as well as between polarity proteins and other interactors (*Chen and Macara, 2005*; *Itoh et al., 2001*; *Médina et al., 2002*; *Michel et al., 2005*; *Roh et al., 2002a*; *Takekuni et al., 2003*; *Tan et al., 2020*). Altogether, these studies provide a complex potential model of molecular interactions. However, in most cases, we do not know to what extent and where these interactions do occur in cells and whether they are transient or permanent. Notably, most of these previous studies used overexpression to identify the interactors of a given protein; this methodological limitation might have introduced biases in some cases. If we focus only on proteins observed in this article, these studies concluded that PAR6β can interact with PAR3, aPKC, PALS1, PATJ, and CRB3A; aPKC can additionally interact with PAR3, and PALS1 can interact with CRB3A and PATJ. Our results show that of these eight potential interactions, only two, namely, PAR6β-aPKC and PALS1-PATJ, are likely to be stable or spatially restricted, clarifying the current view on how polarity proteins interact in the apical junction area of the cell. We question the existence of the canonical Crumbs and PAR complexes as previously described and propose that only PAR6β-aPKC and PALS1-PATJ can be defined as major structural complexes. The other numerous possible interactions that have been claimed previously may exist transiently and our approach cannot rule out that they occur at other locations in the cell, but it questions their relevance to the understanding of the epithelial cell junctions.

The interaction between PAR3, PAR6, and aPKC is key to epithelial polarization (*Horikoshi et al., 2009*; *Joberty et al., 2000*) but the permanence of these interactions has been discussed in the past. In mammalian epithelial cells, PAR3, PAR6, and aPKC have been thought to interact at apical junctions as these proteins concentrate there, but only PAR6 and aPKC are found at the apical surface (*Martin-Belmonte et al., 2007*; *Satohisa et al., 2005*). Moreover, in a few non-mammalian systems, PAR3 was observed as segregated from PAR6 and aPKC at epithelial apical junctions: when observed with confocal microscopy, PAR3 is basal of PAR6 and aPKC in the apical junctions of *Drosophila melanogaster* embryos during cellularization (*Harris and Peifer, 2005*), as well as in chick neuroepithelial cells (*Afonso and Henrique, 2006*). Our data suggest that the segregation of PAR3 from PAR6-aPKC is likely to be a conserved principle of organization in polarized epithelia. Even if the interaction of PAR3 with PAR6-aPKC is central to polarization, it is likely to be transient. The mechanistic basis for the transient character of the interaction between PAR3 and PAR6-aPKC in mammalian epithelia may be similar to the Cdc-42-dependent mechanisms found in *D. melanogaster* (*Morais-de-Sá et al., 2010*) or *C. elegans* (*Rodriguez et al., 2017*; *Wang et al., 2017*). Interestingly, studies in *C. elegans*

also found that polarity proteins can organize in clusters (*Dickinson et al., 2017*; *Wang et al., 2017*). Although this was observed in very different conditions, this points to a potentially conserved mechanism on how polarity proteins organize.

Our finding that PAR3 localizes at the TJ confirms previous observations using electron microscopy in rat small intestine (*Izumi et al., 1998*) and MDCK cells. One recent study found a small fraction of PAR3 at the level of the adherens junction (*Tan et al., 2020*). Even though STED allows for a much larger volume to be probed compared to electron microscopy, we did not observe PAR3 basal of the TJ. The localization of PAR3 may depend on the cell type as well as its maturation state, but interestingly PAR3 is never found in the region apical of the TJ, where we find the other polarity proteins.

Because CRB3 is a transmembrane protein and several studies reported its interaction with PALS1, it was thought to anchor PALS1 and PATJ to the apical membrane (*Makarova et al., 2003*; *Roh et al., 2002b*). Similarly, it is suggested in *D. melanogaster* that Crb recruits Par-6 and aPKC to the apical membrane (*Morais-de-Sá et al., 2010*). Our study suggests that the recruitment of PALS1, PATJ, PAR6β, and aPKC to the plasma membrane is unlikely to be due to CRB3A because CRB3A poorly colocalizes with these proteins. Nevertheless, our data suggest that PALS1-PATJ are localized at the plasma membrane, perhaps confined in this area by another set of interactors to be uncovered. This last observation is likely to be similar for PAR6-aPKC. We cannot rule out both for PALS1-PATJ and PAR6β-aPKC that the interaction with CRB3A could be transient, and that this transient interaction would be sufficient to localize these protein complexes in the apical surface area.

The importance of polarity proteins for the epithelial organization points to the fact that these proteins are likely to play a key role in the organization of the cytoskeleton. Several proteins having a role in actin regulation have been shown to interact with polarity proteins (*Bazellières et al., 2018*; *Médina et al., 2002*), but how polarity proteins could influence actin organization is largely unknown. The correlation of organization between the actin cytoskeleton and PAR6-aPKC and PALS1-PATJ clusters points to a potential structural or instructional role of these proteins in the cytoskeleton organization. These findings call for further investigations, including functional and structural approaches.

We have shown that our observations were not the consequence of methodological artifacts, as live imaging on PAR6β-Citrine CRISPR-Cas9 knock-in and immunolabelling gave very similar results on the same cells, and permeabilization with or without detergent allowed us to observe the same organization of PATJ (*Figure 3*). Nevertheless, because of the size of antibodies, the combination of primary and secondary antibodies displaces the fluorescent signal from the epitope it targets by about 25 nm. Can this displacement alter our conclusions regarding protein organization? The fluorescent densities observed with STED are very likely to be the result of the clustering of tens of fluorophores, therefore tens of antibodies recognizing tens of polarity proteins. Because these densities are created by tens of antibodies, we speculate that these antibodies are oriented isotropically in space. As a result, the use of antibodies might slightly increase the size of polarity protein clusters, but the fluorescence created by antibodies in images is well centered on polarity protein clusters, making antibodies a tool of choice for this study.

Are clusters of polarity proteins smaller than the diffraction limit? It is often not the case, as many fluorescent densities can be as large as twice the diffraction limit and often display complex shapes, indicating clusters of polarity proteins larger than 80 nm. It would be very valuable to understand how these clusters are organized at a smaller scale, and how other proteins are implicated, including cytoskeletal proteins. Moreover, single-molecule approaches used in *Dickinson et al., 2017* could help to further identify the nature of polarity protein clusters and their stoichiometry.

In this study, we define endogenous polarity protein organization and how polarity proteins are likely to interact. The early concept of polarity protein complexes introduced by biochemical studies is impractical today because of the very large number of potential interactions between proteins discovered. Additionally, it omits important features, such as the dynamics of interaction and their reality in relation to cell subregions. Our study proposes a snapshot of the polarity organization in mature intestinal epithelial cells that calls for a novel, more dynamic definition of interactions between polarity proteins and associated proteins that will be needed to uncover the mechanistic basis of cell apico-basal polarization.

# Materials and methods

## Cell culture

A clone of Caco-2 cells, TC7, was used in this study because differentiated TC7 cells form a regular epithelial monolayer (*Chantret et al., 1994*). Caco-2/TC7 cells were kindly provided by Dr. A. Zweibaum (INSERM, U170, Villejuif, France) and were identified as Caco-2 cells by STR profiling (DSMZ ACC 169). Cells were tested negative for mycoplasma with Mycostrip kits (Invivogen). Prior experiments, cells were seeded at a low concentration of $10^5$ cells on a 24 mm polyester filter with 0.4 µm pores (3450, Corning Inc, Corning, NY). Cells were maintained in Dulbecco's modified Eagle's minimum essential medium supplemented with 20% heat-inactivated fetal bovine serum and 1% non-essential amino acids (Gibco, Waltham, MA), and cultured in 10% $CO_2$/90% air. The medium was changed every 48 hr.

## Preparation of the CRISPR-Cas9 PAR6β-Citrine knock-in

The Caco-2 PAR6β Citrine knock-in cell line expressing PAR6β Citrine from the human *PARD6B* locus was prepared by genome editing with the CRISPR-Cas9 method (*Sandoz et al., 2019*). Chemically modified guide RNA (gRNA) and *Streptococcus pyogenes* Cas9 protein were purchased from Synthego. This gRNA targeted the stop codon region of the *PARD6B* exon 3 with this sequence: ATCATAACATTA TGAAACCG (TGG). The donor plasmid was constructed with Citrine coding sequences inserted at the position of the *PAR6DB* STOP codon and flanked by the 5′ homology arm chr20: 50749657–50750486 (*PARD6B*) and 3′ homology arm chr20: 50750962–50750487 (*PARD6B*) (see Appendix 1). The donor plasmid was purchased from Twist Bioscience. The gRNA was diluted following the manufacturer's instruction and ribonucleoprotein complexes were formed with 30 pmoles of gRNA and 12 pmoles of spCas9. $10^6$ Caco-2 cells were transfected with ribonucleoprotein (RNP) and 5 µg donor plasmid using Amaxa Nucleofector (kit T, B24 and T20 programs). Transfected cells were seeded in Petri dishes. After 15 days, fluorescent cells were sorted by a cytometer FACSAriaII and cloned in 96-well plates. After the expansion of the fluorescent clones, the screening of PAR6β Citrine KI cells was performed by Western blot analysis with anti-PAR6β and anti-GFP antibodies (see Appendix 1). The targeted insertion was confirmed by PCR amplification of the expected junction fragments followed by Sanger sequencing (Eurofins). A clone (A6) that expresses PAR6β Citrine but no longer expresses PAR6β (due to the modification of both *PARD6B* gene alleles) was used for further experiments (see Appendix 1).

## Sample preparation for immunostaining

### Human sample preparation

Human intestine biopsies were obtained under the agreement IPC-CNRS-AMU 154736/MB. Intestinal samples were fixed in paraformaldehyde (PFA 32%, Fisher Scientific) 4% in phosphate-buffered saline (PBS, Gibco) for 1 hr at 20°C. Biopsies were embedded in optimal cutting temperature compound (OCT compound, VWR) and frozen in liquid nitrogen.

### Mouse sample preparation

Mouse intestine samples were obtained following local ethical guidelines. After washing with PBS, intestinal samples were fixed in PFA 4% in PBS for 20 min at room temperature. Samples were then embedded in OCT compound and frozen in liquid nitrogen.

### Cell culture preparation for optical microscopy

Cells were washed in PBS and then fixed in PFA 4% in PBS for 20 min at room temperature. When apico-basal orientation observations were needed, cells were sectioned along the apico-basal axis. Before sectioning, cells were embedded in OCT compound and frozen in liquid nitrogen.

### Samples sectioning

When needed, samples were sectioned with a cryostat (Leica CM 3050S, Leica Biosystems, Germany). 10 µm sections were transferred to high-precision 1.5H coverslips (Marienfeld, Germany) previously incubated with poly-L-lysine solution (P-4832, Sigma-Aldrich, St. Louis, MO).

## Immunostaining for optical microscopy

Intestinal sections and cultured cells were prepared similarly. Intestinal sections were permeabilized in 1% SDS (Sigma-Aldrich) in PBS for 10 min. In cultured cells, 10 min of permeabilization was achieved with 1% SDS in PBS for CRB3A antibody, as well as PAR6β and aPKC antibodies when used in combination with TJ markers; otherwise, all other protein labeling protocols included 1% Triton X100 (Sigma-Aldrich) in PBS at permeabilization for 10 min. After washing with PBS, samples were saturated with 10% fetal bovine sera (Gibco) in PBS ('saturation buffer') over an hour at room temperature. Primary antibodies were diluted in the saturation buffer and incubated overnight at 4°C. In more details: rabbit anti-ZO-1 (1/500, 61-7300, Invitrogen), mouse anti-Occludin (1/500, 331500, Invitrogen), mouse anti-E-cadherin (1/500, 610181, BD Biosciences), rabbit anti-PAR3 (1/200, 07-330, Sigma-Aldrich), rabbit anti-PAR6β (1/200, sc-67393, Santa-Cruz), rabbit anti-PKC ζ (1/200, sc-216, SantaCruz), mouse anti-PKC ζ (1/200, sc-17781, SantaCruz), chicken anti-PALS1 (1/200, gift of Jan Wijnholds *Kantardzhieva et al., 2005*), rabbit anti-PATJ (1/200, *Massey-Harroche et al., 2007*; *Michel et al., 2005*), rat anti-CRB3A (1/50 MABT1366, Merck; see antibody validation in Appendix 1). Secondary antibodies were incubated for 1 hr at room temperature. Alexa Fluor 568 conjugated to antibodies raised against mouse, rabbit, and rat, and Alexa Fluor 532 conjugated to antibodies raised against mouse and rabbit (Invitrogen) were used at 1/200 dilution in the saturation media. Phalloidin Alexa Fluor A532 (Invitrogen) was mixed with secondary antibodies and used at 1/100 dilution. After each incubation, samples were rinsed four times with PBS. Samples were finally mounted in Prolong Gold antifade mountant (Invitrogen) at 37°C for 45 min.

## STED microscopy

Images of samples were acquired with a STED microscope (Leica TCS SP8 STED, Leica Microsystems GmbH, Wetzlar, Germany), using a ×100 oil immersion objective (STED WHITE, HC PL APO ×100/1.40, same supplier). Two-color STED was performed with Alexa Fluor 532 excited at 522 nm (fluorescence detection in the 532–555 nm window), and Alexa Fluor 568 excited at 585 nm (fluorescence detection in the 595–646 nm window). To minimize the effect of drifts on imaging, both dyes were imaged sequentially on each line of an image and depleted using the same 660 nm laser. Detection was gated to improve STED signal specificity. The parameter used to generate 3D STED was set to 100% in LASX. To ease the analysis, when imaging with 3D STED, we picked junctions where microvilli were vertical, while in other imaging analyses the orientation of microvilli was unknown because of the lack of actin labeling.

## Live-STED microscopy

Citrine fluorescent proteins were imaged similarly to organic dyes. Citrine was excited at 507 nm, depleted at 592 nm, and fluorescence was detected in the 517–582 nm range. Special care was taken to limit cell damage as we found that too high STED laser excitation led to major cell damage.

## Cultured cell preparation for electron microscopy

Cells were washed in PBS and then fixed in PFA 4% in PBS for 20 min at room temperature. After rinsing with PBS, cells were put into a sucrose gradient to reach 30% sucrose overnight. Cells were then frozen in liquid nitrogen and immediately thawed at room temperature. Immunostaining was carried out without permeabilization step, directly with primary antibodies (rabbit anti-PATJ 1/100, for 3 hr at room temperature). After the washing steps, cells were incubated with a secondary antibody carrying 6 nm gold particles (goat anti-rabbit 1/20, 806.011, Aurion, The Netherlands). A tertiary antibody was used to observe where gold particles were localized on a macroscopic level (Alexa 568 conjugated donkey anti-goat 1/200 from Invitrogen, for 1 hr at room temperature).

Cells were then prepared specifically for electron microscopy. They were fixed in 2.5% glutaraldehyde, 2% PFA, and 0.1% tannic acid in sodium cacodylate 0.1 M solution for 30 min at room temperature. After washing steps, cells were post-fixed in 1% osmium in sodium cacodylate 0.1 M solution for 30 min at room temperature and contrasted in 2% uranyl acetate in water solution for 30 min at room temperature. Cells were then dehydrated in ethanol and embedded in Epon epoxy resin.

## Electron microscopy

Cells were observed with a transmission electron microscope, FEI Tecnai G2 200 kV (FEI, The Netherlands), in an electron tomography mode. Tomograms were reconstructed using the Etomo tool of the IMOD software.

## Data analysis

### Analysis of protein density

To quantify the density and positions of polarity proteins with respect to TJ markers, we used custom-made ImageJ macros and Python programs. In each case, the reference protein was a TJ protein (ZO-1 or Occludin) that was localized precisely, defining a reference position along the junction from which intensity measurement was done. For planar orientations, the reference was the maximum intensity of the TJ marker along the junction; intensity measurements consisted in getting the intensity profiles of proteins perpendicular to the junction, all along the junction. For apico-basal orientations, we measured intensity profiles on the apico-basal axis, all along the junction. On a given profile, the reference was taken at the most apical point where the TJ marker intensity was a third of its maximum intensity; the reason for this choice is that TJs spread along the apico-basal axis tended to vary up to threefold from one cell to another and this definition of the reference allowed us to define a reproducible apical edge of the TJ. In the process, we used bilinear interpolation to obtain subpixel quantification. The results of analyses were then normalized for intensity for each junction to avoid junction-to-junction intensity variation. Because we used a reference protein for each junction, we could then align all results based on the reference position of the reference protein and pool all results into a single protein density plot. Since we used either ZO-1 or Occludin as references, we controlled that these proteins are localized similarly in *Figure 1* and *Figure 2—figure supplement 1*.

### Protein–protein proximity analysis

The principle of quantification of protein–protein proximity was introduced by *Wu et al., 2010*. The authors of this method observed that the autocorrelation of a given image or the cross-correlation between two images coming from two different channels showed a peak at its center. The ratio of amplitude between the peaks of the cross-correlated and autocorrelated images gave a good estimate of protein proximity, which they coined the protein–protein proximity index. This index is similar to more classical colocalization coefficients, but we found that the method of *Wu et al., 2010* was well suited for proteins distributed along a junction.

In practice, we extracted junctions from two-color images, restricting the analysis to a band of 400 nm centered on the reference given by the TJ (as defined earlier). As we found the analysis to be dependent on orientation, when planar orientation was used, we excluded junctions that were not straight. All extracted junctions of a given protein pair to be examined were then concatenated into one large two-channel image on which we achieved autocorrelation and cross-correlation analysis (autocorrelation is achieved on each channel, and cross-correlation is achieved with both channels). We extracted the amplitude of peaks obtained in each of the autocorrelated and cross-correlated images as proposed in *Wu et al., 2010*. Therefore, when analyzing protein 1 and protein 2 proximity, we obtain the amplitude $A_1$ and $A_2$ from the autocorrelation of images of protein 1 and protein 2, respectively, and the amplitude $C_{12}$ from the cross-correlation analysis. One evaluates the fraction of protein 1 colocalizing with protein 2 with the protein–protein proximity index $P_1 = C_{12}/A_2$, and the fraction of protein 2 colocalizing with protein 1 with the protein–protein proximity index $P_2 = C_{12}/A_1$.

In *Figure 4*, we color-coded the values of these indices. To obtain an absolute representation of these values, we additionally used Venn diagrams to represent graphically for each protein the fraction of colocalizing and non-colocalizing protein.

## Acknowledgements

We would like to thank Flora Poizat for the human biopsies, Charlotte Mourry for help during the PAR6β CRISPR-Cas9 knock-in screening, and the Le Bivic and Lenne groups for discussion. Biopsies were obtained with the agreement IPC-CNRS-AMU 154736/MB. We acknowledge the IBDM imaging facility, member of the national infrastructure France-BioImaging supported by the French National Research Agency (ANR-10-INBS-04). PM was supported by ITMO Cancer (Plan Cancer),

Ligue Nationale Contre le Cancer, and the French National Research Agency (ANR-T-JUST, ANR-17-CE14-0032). The project was developed in the context of the LabEx INFORM (ANR-11-LABX-0054) and the A*MIDEX project (ANR-11-IDEX-0001-02), funded by the 'Investissements d'Avenir' French Government program.

## Additional information

### Funding

| Funder | Grant reference number | Author |
|---|---|---|
| Agence Nationale de la Recherche | ANR-10-INBS-04 | Pierre Mangeol |
| Agence Nationale de la Recherche | ANR-17-CE14-0032 | Pierre Mangeol |
| Instituts thematiques multi-organismes | Plan Cancer | Pierre Mangeol |
| Agence Nationale de la Recherche | ANR-11-LABX-0054 | Pierre Mangeol |
| Agence Nationale de la Recherche | ANR-11-IDEX-0001-02 | Pierre Mangeol |
| Ligue Contre le Cancer | Post doctoral grant | Pierre Mangeol |

The funders had no role in study design, data collection and interpretation, or the decision to submit the work for publication.

### Author contributions

Pierre Mangeol, Conceptualization, Resources, Data curation, Software, Formal analysis, Funding acquisition, Investigation, Visualization, Methodology, Writing - original draft, Writing – review and editing; Dominique Massey-Harroche, Methodology, Writing – review and editing; Fabrice Richard, Jean-Paul Concordet, Methodology; Pierre-François Lenne, André Le Bivic, Conceptualization, Supervision, Funding acquisition, Validation, Methodology, Writing – review and editing

### Author ORCIDs

Pierre Mangeol  http://orcid.org/0000-0002-8305-7322
Pierre-François Lenne  http://orcid.org/0000-0003-1066-7506

### Ethics

Human biopsies were obtained with the agreement IPC-CNRS-AMU 154736/MB between the lab of the authors and the Paoli-Calmettes institute (Marseilles, France).

### Decision letter and Author response

Decision letter https://doi.org/10.7554/eLife.62087.sa1
Author response https://doi.org/10.7554/eLife.62087.sa2

## Additional files

### Supplementary files

- Transparent reporting form
- Source code 1. ImageJ marco toolsets to extract intensities from images and Python script to compute cross-correlation used in proximity analysis.

### Data availability

All data generated or analyzed during this study are included in the manuscript and supporting files.

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

## Appendix 1

### Characterization of the rat monoclonal anti-CRB3 antibody

Rat monoclonal anti-CRB3 (obtained by Le Bivic Team after immunization with 15 amino acids from the C-terminal region of human CRB3) is now commercialized by Merck. To characterize this antibody, Caco-2 cells were transfected with siRNA control and siCRB3 by electroporation using Amaxa technology (kit T, B24 program) as described previously in *Vacca et al., 2014*. After 3 days, CRB3 levels of transfected cells were analyzed with the rat monoclonal anti-CRB3 antibody by Western blot assay (n = 3 independent experiments) and immunofluorescence (*Appendix 1—figure 1*).

The siRNA sequences used were: siCT (5′-CGUACGCGGAAUACUUCGAtt-3′, Ambion), siCRB3 (5′-GCAAAUACAGACCACUUCU-3′, 5′-CUGCUAUCAUCGUGGUCUU-3′, 5′-GUGCGGAAGCUU CGGGAGA-3′, 5′-GCUUAAUAGCAGGGAAGAA-3′, Dharmacon [On-Target plus Smart Pool]).

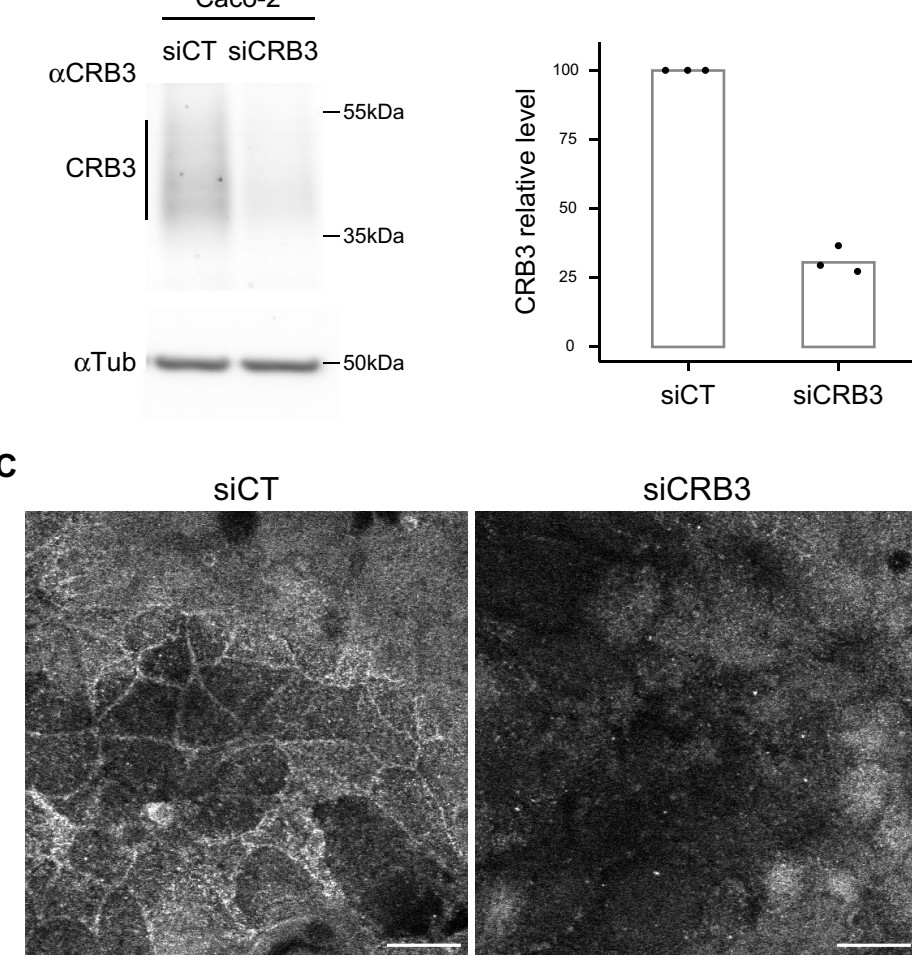

**Appendix 1—figure 1.** Characterization of α-CRB3 antibody. (**A**) Immunoblot analysis of CRB3 expression level in CT (siCT) and CRB3 knock-down (siCRB3) Caco-2 cells with the rat monoclonal α-CRB3 antibody. α-Tubulin is used as a loading control. (**B**) Quantification of CRB3 in siCT and siCRB3 cells. (**C**) Confocal imaging of siCT and siCRB3 Caco-2 cells labeled with the rat monoclonal α-CRB3 antibody. Scale bars: 20 μm.

The online version of this article includes the following source data for appendix 1—figure 1:

**Appendix 1—figure 1—source data 1.** Unnanotated blots in white light, fluorescence and annotated blots.

## Characterization of the CRISPR/Cas9 Caco-2^Par6β::Citrine cells

### A

Gggaggcgttcgggccacagcggata^50749657AGAAGAAGCAGACTACAGTGCCTTTGGTACAGACACGCTAATAAAGAAGAAG
AATGTTTTAACCAACGTATTGCGTCCTGACAACCATAGAAAAAAGCCACATATAGTCATTAGTATGCCCCAAGACTTTAGACC
TGTGTCTTCTATTATAGACGTGGATATTCTCCCAGAAACGCATCGTAGGGTACGTCTTTACAAATACGGCACGGAGAAACCCC
TAGGATTCTACATCCGGGATGGCTCCAGTGTCAGGGTAACACCACATGGCTTAGAAAAGGTTCCAGGGATCTTTATATCCAGG
CTTGTCCCAGGAGGTCTGGCTCAAAGTACAGGACTATTAGCTGTTAATGATGAAGTTTTAGAAGTTAATGGCATAGAAGTTTC
AGGGAAGAGCCTTGATCAAGTAACAGACATGATGATTGCAAATAGCCGTAACCTCATCATAACAGTGAGACCGGCAAACCAGA
GGAATAATGTTGTGAGGAACAGTCGGACTTCTGGCAGTTCCGGTCAGTCTACTGATAACAGCCTTCTTGGCTACCCACAGCAG
ATTGAACCAAGCTTTGAGCCAGAGGATGAAGACAGCGAAGAAGATGACATTATCATTGAAGACAATGGAGTGCCACAGCAGAT
TCCAAAAGCTGTTCCTAATACTGAGAGCCTGGAGTCATTAACACAGATAGAGGCTAAGCTTTGAGTCTGGACAGAATGGCTTTA
TTCCCTCTAATGAAGTGAGCTTAGCAGCCATAGCAAGCAGCTCAAACACGGAATTTGAAACACATGCTCCAGATCAAAAACTC
TTAGAAGAAGATGGAACAATCATAACATTA^50750486*Ggatcc*ATGGTGAGCAAGGGCGAGGAGCTGTTCACCGGGGTGGTGCCC
ATCCTGGTCGAGCTGGACGGCGACGTAAACGGCCACAAGTTCAGCGTGTCCGGCGAGGGCGAGGGCGATGCCACCTACGGCAA
GCTGACCCTGAAGTTCATCTGCACAACAGGAAAACTGCCTGTTCCTTGGCCTACACTTGTTACAACATTTGGATACGGCCTGA
TGTGCTTCGCCCGCTACCCCGACCACATGAAGCAGCACGACTTCTTCAAGTCCGCCATGCCCGAAGGCTACGTCCAGGAGCGC
ACCATCTTCTTCAAGGACGACGGCAACTACAAGACCCGCGCCGAGGTGAAGTTCGAGGGCGACACCCTGGTGAACCGCATCGA
GCTGAAGGGCATCGACTTCAAGGAGGACGGCAACATCCTGGGGCACAAGCTGGAGTACAACTACAACAGCCACAACGTCTATA
TCATGGCCGACAAGCAGAAGAACGGCATCAAGGTGAACTTCAAGATCCGCCACAACATCGAGGACGGCAGCGTGCAGCTTGCT
GATCATTATCAGCAGAATACACCTATTGGAGATGGACCTGTTCTGCTGCCCGACAACCACTACCTGAGCTACCAGTCCGCACT
GAGCAAAGACCCCAACGAGAAGCGCGATCACATGGTCCTGCTGGAGTTCGTGACCGCCGCCGGGATCACTCTCGGCATGGACG
AGCTGTACAAG**TAA**ggatcc**taggtgag**^50750487**tga**AA**t**CG**c**GGTTTGAATGTTTTCAGAGTGAGGATGCCATGAGGACTTGT
ACATTTGGCTAGTTTAAAAGCATATATACCTCTGACCAGTGACGTGGAATAGGCATGAGACGAGTAACGTTGCAAGCTTACAA
TATTATTAAAGTAGTAGTTTGATAATTGTTAATATAAACTTTGGTGGATCAGAGGTGAATTTAAGTCCAAAACAAAGGGGCCT
TTGCTGACGGATTTACGTGCTTTTGCTGTTTTGTCTGTGGAGAATCAGATGTTAAAGCACATTCTTGGAACTATGTGAGAAGA
CTAGATCATTTCTGTTGGAAGTGGTTGCATATTTAACCTGCTGTGCAGAGCCCAGTTAATTTTTCCTTTAACTGTATTTTTAA
AATTCTAATGTGAAGTCTGATTCTCTCTTGTGGTACATTGGGGvACCTCAGCTCTTAAAGGTCTCATGTTCCCAATATTTTATT
TTGATTTTTTTTT^50750962Gcaccgctgtggcccgaacgcctccc

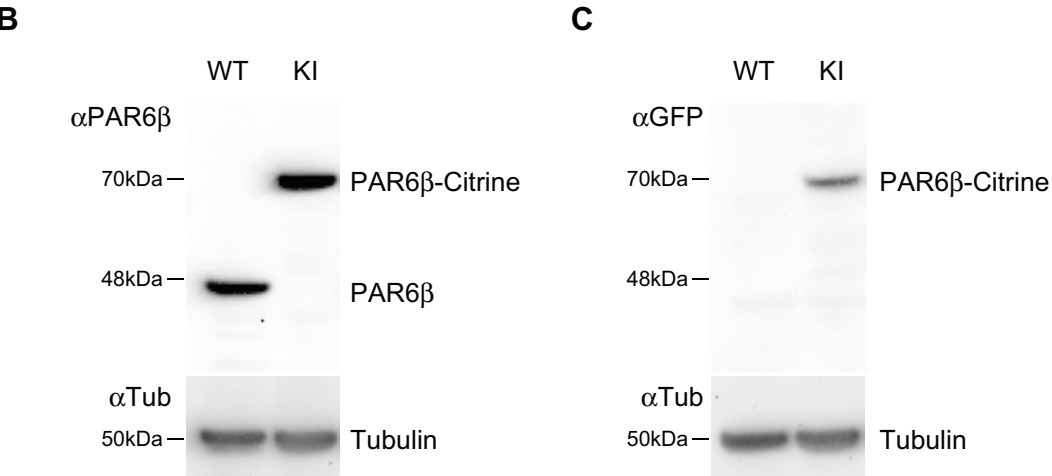

**Appendix 1—figure 2.** Characterization of the CRISPR/Cas9 Caco-2^Par6β::Citrine cells. (**A**) Donor Par6β-Citrine sequence: letters in red, universal guide sequence used by Cas9 to release the plasmid repair matrix into transfected cells; in black, sequence of homology arms; in green, GS peptide linker (and BamHI site to replace the Citrine sequence with another cDNA); in black underlined, Citrine sequence; in bold red, stop codon; in bold violet, a sequence including a STOP codon in each reading phase; in green highlight, sequence matching the gRNA. Mutations (in bold lowercase) were introduced in the noncoding part so that the gPARD6 guide could not lead to a cut (by Cas9) in the donor or modified genomic sequence after insertion. (**B**) Immunoblot analysis of PAR6β expression level in wild-type and Caco-2^Par6β::Citrine Caco-2 cells. α-Tubulin is used as a loading control. (**C**) Immunoblot analysis of PAR6β-Citrine expression level in wild-type and Caco-2^Par6β::Citrine cells Caco-2 cells. B and C deposits were independent.

