## [Editor Report]

This important work advances our understanding of how apical polarity proteins are organized in epithelia using advanced, super-resolution imaging. The microscopy performed in this study is skillful and beautifully presented and clarifies the view of how polarity proteins may interact in mature epithelia. This article will be of interest to cell biologists, especially those interested in cell polarity and tissue architecture.

---

## [Decision Letter]

**Decision letter after peer review:**

Thank you for submitting your article "Super-resolution imaging uncovers the nanoscopic segregation of polarity proteins in epithelia" for consideration by *eLife*. Your article has been reviewed by 3 peer reviewers, including Danelle Devenport as Reviewing Editor and Reviewer #1, and the evaluation has been overseen by Anna Akhmanova as the Senior Editor.

The reviewers have discussed the reviews with one another and the Reviewing Editor has drafted this decision to help you prepare a revised submission.

Summary:

The manuscript addresses a fundamental problem: the organisation of epithelial polarity determinants at the apical domain of human epithelial cells. Mangeol et al. investigate this question using super-resolution microscopy approaches (STED) in polarised intestinal epithelial cells. Using immunofluorescence labeling of endogenous proteins, they provide a careful characterization of Par3-Par6-aPKC and Patj-Pals1-Crb3a localization relative to tight junctions. They find that each protein localizes in the near vicinity of the tight junction, in a clustered organization. Through pairwise colocalization analyses, they observe significant separation of polarity proteins that are generally considered to be part of the same molecular complex based on biochemical assays. Specifically, PAR3 is not associated with aPKC or PAR6, and CRB3a colocalizes poorly with all other polarity proteins, raising interesting questions for future analyses.

The imaging performed in this study is skillful and beautifully presented and, achieving an isotropic resolution of about 80nm, is impressive. However, because of this great gain in resolution compared to other studies of similar components, the major concern of all three reviewers is that the process of fixation and immunostaining may introduce artefacts that are causing the segregated dots to appear. Variable antibody quality and insufficient validation of antibody specificity raise additional concerns about the observed patterns of localization.

Essential Revisions:

The issues with fixation and antibodies would be best addressed by generating endogenously tagged, CRISPR-knockin GFP versions of some of the proteins studied, which is technically straightforward to perform these days, and would allow the conclusions to be drawn with full confidence. As such experiments would constitute major experimental revisions and will take some time to complete, we recommend confirming the most novel and possibly contentious protein pairs: for example PALS1 versus aPKC, and perhaps CRB3a compared to others.

Major points:

Related to fixation and immunofluorescence:

1. All of the results depend on antibody quality, specificity and antigenicity, but no antibody validation provided (with the exception of PATJ). If one primary antibody is less specific than the others, the colocalisation data will be heavily skewed, appearing not to be colocalised. Perhaps this can explain why Crb3a fails to colocalise with the other proteins? Validating the results with a second primary antibody or an endogenously tagged GFP-fusion protein would alleviate this concern.

2. Antibodies are in the range of 10-15nm in length, so with an isotropic resolution of 80 nm, this might have to be taken into account when using primary and secondary antibodies to reveal proteins. In particular, monoclonal versus polyclonal antibodies might have differing effects on localisation precision. I would very much appreciate some comments or thoughts on the fact that polarity proteins were revealed using antibodies.

3. The authors use rather high concentrations of detergent (1% SDS or 1% Triton X-100) for permeabilisation according to their protocols. Are they not worried that this might affect tissue integrity and protein distribution? Couldn't these conditions cause segregated clusters to appear?

Other major points:

1. Figure 1. It would be good to show and demonstrate that Occludin and ZO-1 labeling are completely interchangeable in terms of localisation precision.

2. Figure 3. I do understand the authors' rational for analysing the localisation in the orientation (planar versus apical-basal) that reveals the largest distance, but it would be good to nonetheless show the other orientation for completeness (maybe as supplementary).

3. Figure 5. The final figure shows overlapping PATJ and aPKC at TJs, with the aPKC lost from TJs upon knockdown of PATJ. This result seems to suggest that the CRB-PALS-PATJ complex must directly bind/recruit aPKC to TJs, which seems to conflict with their earlier conclusions that these complexes are completely separated in punctae and do not truly overlap at the nanoscale.

4. Figure 5. The super-resolution characterization of actin organization is not as extensive or convincing as the description of polarity protein localization. A closer examination of actin organization relative to PATJ and aPKC at junctional, apical, and villi positions would strengthen the findings in Figure 5.

5. I am also a bit confused by the analysis presented in Figure 5 with regards to colocalisation of components with apical F-actin structures and the deduction from these and the EM data that some components, aPKC/Par6, localise to 'the first row of' microvilli near junctions whilst PALS1-PATJ localise near the base of said microvilli. How would localisation to the apical plasma membrane outside of or within microvilli be restricted to only the ones near junctions? There is not only F-actin in microvilli but also all over and near the apical cortex, so what distinguished the ability of aPKC/PAR6 to bind to actin in microvilli? The PATJ knock-down results are interesting, and I agree suggestive of some interaction between the complexes and actin organisation. But without further analyses as to what other components might be affected in their localisation in this situation, it is hard to judge whether the effect on actin is a direct or rather indirect one, so I am unsure as to what these images add without more in depth follow-up.

6. In some cases the number of biological replicates is small. Only one mouse sample was used, and the quantifications of junctions are performed across just 1 or 2 cell culture replicates (although more replicates were performed, just not used for quantification). Therefore, the data reflect the variability across junctions (violin plots in Figures1-2) but they don't reflect the variability across biological replicates. This also means the p-value in Figure 5 was calculated using n=number of junctions rather than n=experimental replicates, which would be a more appropriate comparison of means. Quantifying the data across 3 biological replicates to show the variability across experiments would greatly strengthen the results and conclusions.

7. The authors rightly point out where their study fits within what has been attempted by other labs previously in order to understand and dissect apical polarity complex function. They clearly define interesting aspects, such as PALS1-PATJ and aPKC-PAR6 forming independent clusters, and the lack of colocalisation and thus maybe association with Crumbs3. In contrast to the last sentence statement of their abstract 'This organization at the nanoscale level significantly simplifies our view on how polarity proteins could cooperate to drive and maintain cell polarity.' I cannot yet see what these results simplify about our understanding of apical polarity complexes and even more so what the authors' new model is of how the complexes work. This needs to be spelt out more clearly, please.

8. I would also point out that, in part, other studies have pointed in the same direction. The recent paper by the Ludwig lab (Tan et al. 2020 Current Biology 30, 2791-2804) points in part in a similar direction, identifying a vertebrate 'marginal zone' similar to the one already known from invertebrate epithelia, as well as identifying basal to this an apical and basal tight junction area. Furthermore, as the authors themselves discuss in the discussion, the 'splitting away' of Par3 has been observed in *Drosophila* epithelia (embryonic, follicle cells and eye disc), and should maybe be introduced already at an earlier point of the paper. Furthermore, papers by Wang et al. and Dickinson et al., that also analyse PAR complex clustering should be cited and mentioned in the introduction/discussion (Wang, S.-C., Low, T. Y. F., Nishimura, Y., Gole, L., Yu, W., and Motegi, F. (2017). Cortical forces and CDC-42 control clustering of PAR proteins for *Caenorhabditis elegans* embryonic polarization. Nature Cell Biology, 19(8), 988-995. http://doi.org/10.1016/S0960-9822(99)80042-6; Dickinson, D. J., Schwager, F., Pintard, L., Gotta, M., and Goldstein, B. (2017). A Single-Cell Biochemistry Approach Reveals PAR Complex Dynamics during Cell Polarization, 1-42. http://doi.org/10.1016/j.devcel.2017.07.024).

---

## [Author Response]

Essential Revisions:The issues with fixation and antibodies would be best addressed by generating endogenously tagged, CRISPR-knockin GFP versions of some of the proteins studied, which is technically straightforward to perform these days, and would allow the conclusions to be drawn with full confidence. As such experiments would constitute major experimental revisions and will take some time to complete, we recommend confirming the most novel and possibly contentious protein pairs: for example PALS1 versus aPKC, and perhaps CRB3a compared to others.

We agree with the reviewers’ comment that labeling proteins with antibodies can lead to artifacts. As this was reported many times, such artifacts can be due to a lack of specificity of antibodies, or because of fixation or permeabilization. The use of CRISPR knock-in fluorescent versions of the proteins studied can alleviate these artifacts.

We tried to prepare PAR6β, PATJ, and CRB3A with fluorescent tags; PALS1 and aPKC could not be used because they come in several isoforms (Kamberov et al., JBC, 2000; Shaha et al., Placenta, 2022). The CRISPR knock-in of the fluorescent protein Citrine in PAR6β in Caco-2 cells was successful. The GFPtagged version of PATJ we tried resulted in the diffused localization of the protein in the cytoplasm in Caco-2 cells, suggesting artifacts of tagging. Our lab has tried to obtain CRISPR knock-ins of CRB3A in Caco-2 cells without, so far, any success. CRISPR knock-ins are powerful tools, but we can witness as many other labs working on human cell lines that successful knock-ins are strongly protein-dependent and lead to many failures.

To best address the reviewers’ concerns, we designed the following experiment using the CRISPR knock-in PAR6β-Citrine in Caco-2 cells. Since we observed the organization of polarity proteins in fixed samples thanks to superresolution microscopy, the imaging of the fluorescent tag had to be achieved on a live sample with superresolution microscopy to allow for comparison. Moreover, to ensure that we observe the same organization in fixed and live samples, we first imaged with STED the fluorescent protein Citrine in live Caco-2 cells, then fixed, permeabilized, and labeled the cells with antibodies and finally came back to the exact same cells observed in live and imaged them with STED, this time thanks to an AlexaFluor-568 labeled antibody. We find the same clusters of PAR6β when observed live with Citrine or fixed with AlexaFluor-568. This comparison rules out the possibility that the observed clusters are artifactual and confirms our results using superresolution.

How we modified our manuscript:

Examples of these experiments are reported in Figure 3A and we described these results in the main text (lines 131-146, page 9).

Major points:Related to fixation and immunofluorescence:1. All of the results depend on antibody quality, specificity and antigenicity, but no antibody validation provided (with the exception of PATJ). If one primary antibody is less specific than the others, the colocalisation data will be heavily skewed, appearing not to be colocalised. Perhaps this can explain why Crb3a fails to colocalise with the other proteins? Validating the results with a second primary antibody or an endogenously tagged GFP-fusion protein would alleviate this concern.

Of all the antibodies we used, only the characterization of the rat antibody -CRB3A has not been published.

How we modified our manuscript:

A section of the supplementary material is dedicated to the validation of the -CRB3A antibody.

2. Antibodies are in the range of 10-15nm in length, so with an isotropic resolution of 80 nm, this might have to be taken into account when using primary and secondary antibodies to reveal proteins. In particular, monoclonal versus polyclonal antibodies might have differing effects on localisation precision. I would very much appreciate some comments or thoughts on the fact that polarity proteins were revealed using antibodies.

We agree with the reviewers’ comment that when trying to describe how polarity proteins are organized, it is important to consider how antibodies are located with respect to their epitope, how their size can affect the localizations we witness, and how the spatial resolution of the microscope can affect the perceived distribution of polarity proteins.

How we modified our manuscript:

To discuss these points, which question how polarity proteins are organized, we added the following text in the discussion part of our manuscript (lines 364-374, page 23).

“We have shown that our observations were not the consequence of methodological artifacts, as live imaging on PAR6β-Citrine CRISPR-Cas9 knock-in and immunolabelling gave very similar results on the same cells, and permeabilization with or without detergent allowed us to observe the same organization of PATJ (Figure 3). Nevertheless, because of the size of antibodies, the combination of primary and secondary antibodies displaces the fluorescent signal from the epitope it targets by about 25nm. Can this displacement alter our conclusions regarding protein organization? The fluorescent densities observed with STED are very likely to be the result of the clustering of tens of fluorophores, therefore tens of antibodies recognizing tens of polarity proteins. Because these densities are created by tens of antibodies, we speculate that these antibodies are oriented isotropically in space. As a result, the use of antibodies might slightly increase the size of polarity protein clusters, but the fluorescence created by antibodies in images is well centered on polarity protein clusters, making antibodies a tool of choice for this study.”

To keep the discussion on monoclonal *versus* polyclonal antibodies here, the epitopes used to obtain polyclonal antibodies are very small in fully folded proteins, domains encompassing epitopes being in the order of a few nanometers. Therefore, while monoclonal antibodies will always reach the same location in a protein, polyclonal will reach positions that will differ by a few nanometers at most. We believe this effect is negligible in comparison to the use of primary and secondary antibodies.

3. The authors use rather high concentrations of detergent (1% SDS or 1% Triton X-100) for permeabilisation according to their protocols. Are they not worried that this might affect tissue integrity and protein distribution? Couldn't these conditions cause segregated clusters to appear?

We agree with the reviewers’ comment that detergent used for permeabilization could lead to artifacts. To the best of our knowledge, the concentrations used in our manuscript are commonly reported in the field, and we adjusted the concentration to their minimum to obtain reproducible results. To verify that the detergents we use are not responsible for the clustered organization of proteins, we created a protocol where permeabilization with detergents is replaced by freezing and thawing cells once in liquid nitrogen, which partially cracks cell membranes. We already used this protocol in the initial version of the manuscript Figure 4 when deciphering the location of PATJ with electron microscopy. The morphology of cells was preserved in these conditions.

Using this protocol for STED microscopy on PATJ, we could not witness any significant difference with the images obtained with detergent. It is to be noted that only labeling with PATJ worked in these conditions. Freezing and thawing cells was not sufficient to detect any antibody of the other polarity proteins.

The results come in addition to what we observed with the CRISPR knock-in (See our response above 2.) and confirm that our observations are not due to artifacts.

How we modified our manuscript:

Examples of these experiments are reported in Figure 3B and we described these results in the main text (lines 154-159, page 11).

Other major points:1. Figure 1. It would be good to show and demonstrate that Occludin and ZO-1 labeling are completely interchangeable in terms of localisation precision.

We agree with the reviewers’ comment that this is important to show. We find that the maxima of Occludin and ZO-1 along the apico-basal axis are distant by at most 15 nm, which is sufficient to be used as interchangeable references.

How we modified our manuscript:

The revised Figures 1 and 2 and their supplemental Figure 1 now display examples of the co-labeling of Occludin and ZO-1 as well as their analyses.

2. Figure 3. I do understand the authors' rational for analysing the localisation in the orientation (planar versus apical-basal) that reveals the largest distance, but it would be good to nonetheless show the other orientation for completeness (maybe as supplementary).

We agree with the reviewers’ comment that showing both orientations will strengthen our analysis.

How we modified our manuscript:

The corresponding figure (now Figure 4) comes with a supplementary figure displaying perpendicular orientations.

3. Figure 5. The final figure shows overlapping PATJ and aPKC at TJs, with the aPKC lost from TJs upon knockdown of PATJ. This result seems to suggest that the CRB-PALS-PATJ complex must directly bind/recruit aPKC to TJs, which seems to conflict with their earlier conclusions that these complexes are completely separated in punctae and do not truly overlap at the nanoscale.

We now have deleted this part of the figure because we think it is too preliminary and does not bring enough to the current manuscript. A much more extended study would be required to tackle how polarity proteins recruit one another and how they affect the cytoskeleton.

4. Figure 5. The super-resolution characterization of actin organization is not as extensive or convincing as the description of polarity protein localization. A closer examination of actin organization relative to PATJ and aPKC at junctional, apical, and villi positions would strengthen the findings in Figure 5.

We agree with the reviewers’ comment that the characterization of actin organization was not as extensive as the description of polarity protein localization and that the data and analysis provided were insufficient. We devised to describe as precisely as possible how PATJ, PALS1, aPKC, and PAR6β proteins are organized versus actin. As actin is almost entirely organized into microvilli in the region, we described how these proteins are organized versus microvilli. This analysis allowed us to find that PATJ and PALS1 are mostly located at the base and in between microvilli whereas aPKC and PAR6β are mostly located all along microvilli.

How we modified our manuscript:

These experiments are reported in the new Figure 6 and we described these results in the main text (lines 261-275, page 17).

5. I am also a bit confused by the analysis presented in Figure 5 with regards to colocalisation of components with apical F-actin structures and the deduction from these and the EM data that some components, aPKC/Par6, localise to 'the first row of' microvilli near junctions whilst PALS1-PATJ localise near the base of said microvilli. How would localisation to the apical plasma membrane outside of or within microvilli be restricted to only the ones near junctions? There is not only F-actin in microvilli but also all over and near the apical cortex, so what distinguished the ability of aPKC/PAR6 to bind to actin in microvilli? The PATJ knock-down results are interesting, and I agree suggestive of some interaction between the complexes and actin organisation. But without further analyses as to what other components might be affected in their localisation in this situation, it is hard to judge whether the effect on actin is a direct or rather indirect one, so I am unsure as to what these images add without more in depth follow-up.

We believe these concerns are now answered in the new version of Figure 6 as we provide much stronger data and analysis to prove these affirmations. We are also surprised that polarity proteins cluster at the base or in the microvilli closest to the junction. “Why are these microvilli special” remains an observation that will interesting to dissect in future studies.

6. In some cases the number of biological replicates is small. Only one mouse sample was used, and the quantifications of junctions are performed across just 1 or 2 cell culture replicates (although more replicates were performed, just not used for quantification). Therefore, the data reflect the variability across junctions (violin plots in Figures1-2) but they don't reflect the variability across biological replicates. This also means the p-value in Figure 5 was calculated using n=number of junctions rather than n=experimental replicates, which would be a more appropriate comparison of means. Quantifying the data across 3 biological replicates to show the variability across experiments would greatly strengthen the results and conclusions.

We agree with the reviewers’ comment that this needed to be addressed. Data presented in the main figures 1, 2, 4, and 6 are now the result of three replicates, either from cell culture replicates or subjects. The proximity analysis presented in Figure supplement 1 of Figure 4 is obtained from single cell culture as these are very difficult experiments that are confirmations of results obtained in Figures 1 and 2. The results obtained with electron tomography are also obtained with one cell culture replicates, although on several cells.

7. The authors rightly point out where their study fits within what has been attempted by other labs previously in order to understand and dissect apical polarity complex function. They clearly define interesting aspects, such as PALS1-PATJ and aPKC-PAR6 forming independent clusters, and the lack of colocalisation and thus maybe association with Crumbs3. In contrast to the last sentence statement of their abstract 'This organization at the nanoscale level significantly simplifies our view on how polarity proteins could cooperate to drive and maintain cell polarity.' I cannot yet see what these results simplify about our understanding of apical polarity complexes and even more so what the authors' new model is of how the complexes work. This needs to be spelt out more clearly, please.

The main question many previous studies focused on was “what interactions between polarity proteins and other proteins exist”. If we focus only on proteins observed in this manuscript, these studies concluded that PAR6β can interact with PAR3, aPKC, and CRB3A, aPKC can interact with PAR3 and PAR6, PALS1 can interact with CRB3A, PATJ, and PAR6. This has led to multiple reviews and articles, including ours, presenting polarity proteins as a block of many interacting proteins, without any hierarchy in what could be the most relevant interactions.

We show in our manuscript that of these 8 potential interactions between key proteins, only two, namely the pairs PAR6β/aPKC and PATJ/PALS1, are likely to be described as prominent or stable, discarding the other 6 as stable in this area of the cell.

How we modified our manuscript in an attempt to better explain these facts:

– We rephrase the last sentence of the abstract: “Of the numerous potential interactions identified between polarity proteins, only PALS1-PATJ and aPKC-PAR6β are spatially relevant in the junctional area of mature epithelial cells, simplifying our view of how polarity proteins could cooperate to drive and maintain cell polarity.”

– We revised the paragraph addressing this point in the previous version of the manuscript (discussion, pages 20-21, lines 304-323).

8. I would also point out that, in part, other studies have pointed in the same direction. The recent paper by the Ludwig lab (Tan et al. 2020 Current Biology 30, 2791-2804) points in part in a similar direction, identifying a vertebrate 'marginal zone' similar to the one already known from invertebrate epithelia, as well as identifying basal to this an apical and basal tight junction area. Furthermore, as the authors themselves discuss in the discussion, the 'splitting away' of Par3 has been observed in Drosophila epithelia (embryonic, follicle cells and eye disc), and should maybe be introduced already at an earlier point of the paper. Furthermore, papers by Wang et al. and Dickinson et al., that also analyse PAR complex clustering should be cited and mentioned in the introduction/discussion (Wang, S.-C., Low, T. Y. F., Nishimura, Y., Gole, L., Yu, W., and Motegi, F. (2017). Cortical forces and CDC-42 control clustering of PAR proteins for *Caenorhabditis elegans* embryonic polarization. Nature Cell Biology, 19(8), 988-995. http://doi.org/10.1016/S0960-9822(99)80042-6; Dickinson, D. J., Schwager, F., Pintard, L., Gotta, M., and Goldstein, B. (2017). A Single-Cell Biochemistry Approach Reveals PAR Complex Dynamics during Cell Polarization, 1-42. http://doi.org/10.1016/j.devcel.2017.07.024).

We agree with the reviewers’ comment that these references are needed. They are now included in the manuscript.

We did not move the reference to *Drosophila* epithelia earlier in the text (or others) as it felt more straightforward to us to introduce results in the literature directly relating to our models, then describe our findings and finally focus on what relates to our findings in other model systems, potentially pointing at conserved mechanisms.